# Pro-Inflammatory Cytokine Gene Expression in Penile Cancer: Preliminary Studies

**DOI:** 10.3390/medicina59091623

**Published:** 2023-09-08

**Authors:** Mateusz Czajkowski, Piotr M. Wierzbicki, Anna Kotulak-Chrząszcz, Bartosz Małkiewicz, Roman Sosnowski, Zbigniew Kmieć, Marcin Matuszewski

**Affiliations:** 1Department of Urology, Medical University of Gdańsk, 80-214 Gdańsk, Poland; 2Department of Histology, Medical University of Gdańsk, 80-214 Gdańsk, Poland; 3Department of Minimally Invasive and Robotic Urology, University Center of Excellence in Urology, Wrocław Medical University, 50-367 Wrocław, Poland; 4Department of Uro-Oncology, M. Sklodowska-Curie Memorial Cancer Center and Institute of Oncology, 00-001 Warsaw, Poland

**Keywords:** IL-1, IL-1RN, IL-6, IFN-γ, penile cancer, TGF-β1

## Abstract

*Background and Objectives:* Penile cancer is a rare neoplasm in developed countries with an incidence of 0.8/100,000 per male inhabitant. Despite the development of personalized medicine and multimodal treatment, the outcome of penile cancer treatment is insufficient. Our study aimed to assess the levels of pro-inflammatory cytokines’ mRNA such as interleukin 1-A (encoded by *IL1A* gene, alias *IL-1A)*, interleukin 1-B (*IL1B*, *IL-1B*), interleukin 1 receptor antagonist (*IL1RN*, *IL-1RN*), interleukin 6 (*IL6*, *IL-6*), transforming growth factor β1 (*TGFB1*, *TGFβ-1*), and Interferon-gamma (*INFG*, *INF-γ*) in penile cancer tissue and associate them with tumor progression and patient survival. *Material and Methods:* Skin biopsies from patients suffering from penile cancer (n = 6) and unchanged foreskin from 13 healthy adult males undergoing circumcision due to a short frenulum were obtained. Pro-inflammatory cytokine mRNA levels were quantified through qPCR. *Results:* We observed higher expression of pro-inflammatory cytokine genes (IL-1A, IL-1B, IL-6, INF-γ, TGF-β) in penile cancer tissue. The average follow-up period was 48 months (range: 38–54 months), during which only one penile tumor progression was observed However, this was without association with the nature of tumor (patient refused radical treatment). *Conclusions:* This is the first study to show increased expression of cytokines such as IL-1A, IL-1B, IL-6, INF-γ, and TGF-β in penile cancer with positive correlation between TNM staging and INF-γ levels in tumor samples (rs = 0.672, *p* = 0.045), which may be associated with the immunosuppressive role of the tumor environment.

## 1. Introduction

Penile cancer (PC) is a rare neoplasm in western European countries, and the incidence has been stable for three decades, amounting to 0.8/100,000 male inhabitants. Phimosis, smoking history, and high-risk human papillomavirus (HPV) infection are the most common postulated risk factors [1].

In the literature, there is a lack of information about possible biomarkers that could be used to estimate progression in penile cancer. Squamous cell carcinoma (SCC) is the most common histopathological feature in penile cancer. The same histopathologic results were observed in head and neck cancer, where higher expression of interleukin 1-A (encoded by *IL1A* gene, alias *IL-1A*) is associated with metastasis and the worst outcome [2,3].

The tumor immune microenvironment (TIME) appears to be an important factor in the pathogenesis of penile cancer. According to Chu et al., this very complex microenvironment can lead to penile tumor progression [4]. TIME is divided into three types: immune inflamed, immune desert, and immune excluded [5]. Since phimosis and balanitis are considered risk factors for penile cancer, should penile tumors be recognized as inflamed tumors? The immune-inflamed environment is more favorable for T-cell activation and expansion, including *IL-1A, IL-1B, IL-1RN, TGFβ-1,* and INF-*γ* [5]. Moreover, in a recent study, we confirmed the higher expression of NF-κB1 and NF-κB2 (nuclear factor of kappa light polypeptide gene enhancer in B-cells 1 and 2) in penile cancer tissue. NF-κB1 and NF-κB2 take part in stimulating the secretion of proinflammatory cytokines [6]. Additionally, pro-inflammatory cytokines are overexpressed in penile lichen sclerosus, which is considered as a risk factor for penile cancer [7].

Due to these facts, our study aims to assess the mRNA expression of cytokines such as interleukin 1-A, interleukin 1-B (*IL1B, IL-1B*), interleukin 1 receptor antagonist (*IL1RN, IL-1RN*), interleukin 6 (*IL6, IL-6*), transforming growth factor β1 (*TGFB1, TGFβ-1*), and Interferon-gamma (*INFG, INF-γ*) in penile cancer in comparison to healthy foreskins. Moreover, we tried to correlate gene expression with penile cancer progression, recurrence, and the patient’s survival.

## 2. Materials and Methods

### 2.1. Patients and Skin Biopsies

Biopsies from penile tumors were obtained from 6 patients who underwent penile-sparing surgery due to penile cancer at the tertiary referral Department of Urology, between January 2017 and December 2019. The external control group consisted of foreskin specimens obtained from patients who underwent male circumcision where histological examination of whole specimens confirmed normal healthy skin. None of the control patients suffered from local penile, especially foreskin, inflammation. Additionally, C-reactive protein (CRP) levels in venous blood were measured before surgery from each patient. An independent Bioethics Committee approved the present study (decision No. NKBBN/369/2017), and all patients were informed about the project’s details and had signed written informed consent forms before surgery.

### 2.2. Skin Biopsy Acquisition

All biopsies were obtained during procedures such as penile tumor resection or complete circumcision. The indications for surgery were penile cancer (PC, n = 6; 31.6%) and a short frenulum (control group, n = 13; 68.4%). Patients had not received any topical treatment, especially corticosteroids or imiquimod, six months before circumcision. After circumcision, each tissue fragment (tumor for PC patients or foreskin for control patients) was cut into two similar fragments; one was immediately placed in 5 volumes of RNA-Later (Ambion Inc., a brand of Thermo Fisher Scientific, Inc., Waltham, MA, USA), and stored in a fridge for 6 to 24 h, followed by storage in −80 °C until further processing (RNA extraction). The remaining tissue fragment underwent fixation in ~10 volumes of buffered 4% formaldehyde (pH = 7.4, POCH, Gliwice, Poland) and was stored at 4 °C. Formalin-stored tissues were further processed for histopathological assessment.

### 2.3. Assessment of the mRNA Expression of IL-1A, IL-1B, IL-1RN, IL-6, TGF-β1, and IFN-γ Genes

RNA isolation was optimized by adapting a modified method of Chomczynski and Sacchi [8] using a Total RNA Mini protocol isolation kit (A&A Biotechnology, Gdansk, Poland). Briefly, RNA-Later samples were defrosted and drained of liquid with a sterile paper towel, then a 3 × 3 × 3 mm tissue fragment was cut out for RNA extraction. The remaining tissue sample of similar size was placed in a sterile vial and immediately placed in liquid nitrogen, and then was stored at −80 °C. The processed biopsy tissue was cut with sterile scissors into fragments that were as small as possible and placed in a 1.5 mL Eppendorf tube with 800 µL Fenozol. The tube was incubated in TS-100C (BioSan, Rīga, Latvia) thermoblock at 50 °C for 45 min. After adding 200 µL chloroform (POCH), samples were gently mixed and incubated at room temperature (RT) for 5 min, followed by centrifugation at 12,000 rpm for 15 min at 4 °C. The next steps of RNA extraction were carried out following the manufacturer’s protocol with the final elution volume of 100 µL RNAse-free water. After RNA quantity and purity assessment (Epoch 800 plate reader), RNA was stored at −80 °C for further analyses. cDNA synthesis was performed as previously described [9]. Total RNA samples (2 µg) were reverse-transcribed with RevertAid Reverse Transcriptase (Fermentas; Thermo Fischer Scientific, Inc.). Details concerning the qPCR methodology are provided in Table 1. Finally, 1 µL of four-times diluted cDNA was used in 10 µL total volume of qPCR reaction. All reactions were run in duplicate; the measurement of glucuronidase beta (GUSB) gene expression was used for the normalization of qPCR results with Livak and Schmittgen’s 2ΔΔCq method [10,11].

### 2.4. Statistical Analysis

Statistical analysis was performed using GraphPad Prism version 6.07 (GraphPad Software) software. The following statistical tests were used: 2 × 2 Fisher’s exact, Shapiro–Wilk normality; non-parametric Mann–Whitney U, Wilcoxon signed-rank, and Spearman’s correlation. A two-sided *p* < 0.05 was considered to indicate a statistically significant difference, with a 95% confidence interval in all analyses.

## 3. Results

### 3.1. Patient Characteristics

The histopathological examination confirmed clinical suspicion, and based on this, patients were divided into two groups: penile cancer (n = 6; 31.6%) and short frenulum (n = 13; 68.4%).

Pathological stagings (TNM) of penile cancer were pT1aNxMx (n = 3), pT2N0Mx (n = 1), pTiS (n = 1), and carcinoma planoepitheliale veruccosum (n = 1). There were pathologist-characterized penile cancer specimens as well differentiated G1 (n = 5) and moderate differentiated G2 (n = 1).

All patients’ data (demographic, clinical and histopathological diagnosis, follow-up) were recorded in the database. We noted differences in age and CRP concentration values between the control and Ca penis patients (Table 2).

The average follow-up period was 48 months (range: 38–54 months). Only one patient from the penile cancer group (n = 6) had local tumor progression. However, the tumor progression resulted from the fact that the patient refused radical treatment. The rest of the patients had no signs of local and distant recurrence.

### 3.2. Expression of the Cytokine Genes in All Groups of Patients

We observed an increase in the level of expression of all analyzed cytokines in the tumor tissue in relation to the control foreskins. The highest levels were noted for IL-1B and IL-1A (ca. 385 and 165 times, respectively, Figure 1) in penile cancer vs. control group. The mRNA levels of other pro-inflammatory cytokines, IFN-γ and IL-6, were also elevated (ca. 82 and 16 times, respectively, Figure 1). The smallest level increase was observed for TGFβ1, ca. 5 times higher than in the controls.

### 3.3. Cytokine Levels Regarding Clinical–Histological and Outcome Status of Patients

Despite the observed differences in the level of cytokines in the neoplastic tissue, we did not notice any relationship between cytokine expression and age, BMI values, or CRP levels, as presented in Figure 2A–C. However, we found a positive correlation between increasing clinical staging (carcinoma planoepitheliale veruccosum→ TiS→ pT1aNxMx → pT2N0Mx) and IFN-γ mRNA levels in tumor tissue (Figure 2D). There was no association between histopathological grade and cytokine expression (Figure 2E).

## 4. Discussion

Interleukin 1 (IL-1A and IL-1B) is produced by macrophages, keratinocytes, and endothelial cells, which take part in inflammatory or fibrogenic processes [3]. IL-1RN has an antagonistic effect on IL-1 that consists of binding the IL-1α and IL-1β to IL-1RN [12,13]. Another pro-inflammatory cytokine is interleukin 6, which has pleiotropic activity since it is secreted as an acute-phase protein by macrophages similarly to IL-1 [14]. TGF-β1 shows very strong fibrotic action and takes part in wound healing [15]. The recruitment of macrophages depends on INF-γ—an antiviral and proinflammatory cytokine [16]. The interaction between cytokines is responsible for the balance of many processes in the human body, and could take part in carcinogenesis or cancer progression.

For the first time in the literature, we observed increased mRNA levels of all the studied cytokines, except for IL1-RN, in tumor samples of penile squamous cell carcinoma in comparison to control foreskin samples of healthy males. High mRNA and protein levels of IL-1A were observed in head and neck squamous cell carcinoma by Leon et al., who found that high tumor IL-1A mRNA levels were associated with metastasis and the worst outcome [3]. Increased levels of IL-1B mRNA and proteins have been found in various types of cancer [17]. The pro-inflammatory role of IL-1B was connected with the increased invasive potential of chemical-carcinogen-induced tumors in a murine model [18]. IL-1RN blocks receptors for IL-1 family cytokines, and its quantification was the first in penile cancer and one of few in all malignancies [13]. Most authors focus on gene polymorphism and the number of variable number tandem repeats (VNTRs) in intron 2 of IL-1RN [12]. Cauci et al. found a higher frequency of *IL-1RN* VNTR in a group of 133 melanoma cases [13]. Since there is no direct relationship between observed polymorphism and IL-1RN level, our results are the first that show the possible lack of IL-1RN blockage of the local inflammatory function of Il-1A and IL-1B in penile cancer.

Local tumor overexpression of IL-6 has been observed in colorectal, prostate, pancreatic, lung, cervical, and breast cancer as well as in renal and ovarian carcinomas, confirming the important role of this cytokine in tumor survival and progression [14]. The anti-cancer effect of INF-γ is widely known and led to its usage in clinical treatment. However, recent data suggest its suppressive role in tumor immune evasion. Despite the fact that cancer cells cannot secrete INF-γ while NK, CD8+, and Th22 IFN-γ+CD4+ T cells, macrophages, IFN-producing killer dendritic cells (IKDC), and group 1 innate lymphoid cells (ILC1) secrete this cytokine in the tumor environment, complex cross-talk between cancer cells and IFN-γ-producing cells may promote immunosuppressive tumor microenvironments [16]. Our observation of an increase in the expression of the mRNA of the IFN-γ gene with the clinical advancement of the neoplastic disease may confirm the increased immunosuppressive effect of the stromal and influx cells of the immune system. Activation of TGF-β1 expression in tumor cells has been shown to be associated with the acquisition of epithelial-to-mesenchymal transformation ability by those cells in various cancers [19].

Our study revealed the histological structure of penile cancer with a massive presence of inflammatory cells separated by nests formed by cancer cells, which suggests that these cells were mainly responsible for the highly increased cytokine gene expression in penile cancer, which is in line with similar findings in other types of cancer (Figure 1).

Our research demonstrates altered cytokine gene expression without any association with progression and patient survival. However, it should be kept in mind that this may be due to the enrollment of study patients with a low level of penile cancer advancement.

The main limitation of this study is the small number of cases. Moreover, we investigated the expression of genes but not molecules. Future studies are needed for the expression of proinflammatory cytokines and linking their expression with penile cancer progression and patient survival, especially for INF-γ.

## 5. Conclusions

The expression of cytokines such as IL-1A, IL-1B, IL-6, INF-γ, and TGF-β is higher in penile cancer, but it is not associated with penile cancer progression or patients survival, while a positive correlation between INF-γ levels in cancer samples and clinical advancement may be associated with the immunosuppressive role of the tumor environment.

## Figures and Tables

**Figure 1 medicina-59-01623-f001:**
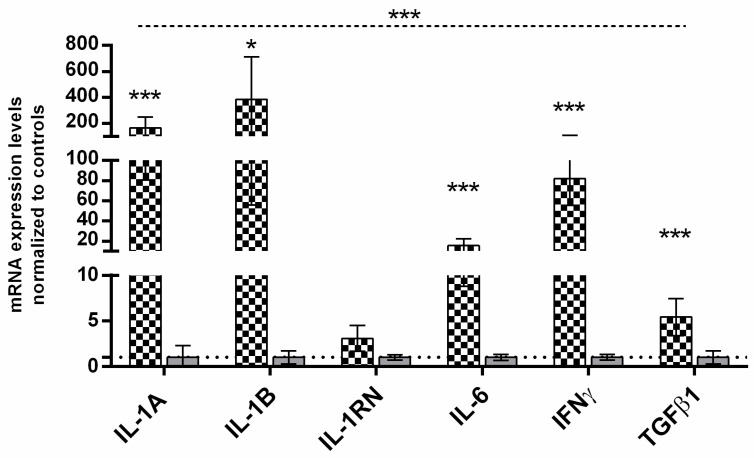
Cytokine gene expression at the mRNA level in penile tissue. Gene expression was assessed as described in the Materials and Methods section. The ordinate axis is divided into three sections to show all important values. Bars and whiskers represent the mean ± standard deviation of the mean (SEM) of cancer tissues (bars with chessboard pattern) normalized to control foreskin samples (gray bars), counted as 1, shown as a dotted horizontal line. * *p* < 0.05, *** *p* < 0.001 (Mann–Whitney U test between each group, Kruskal–Wallis ANOVA test between all groups, upper dotted line). Abbreviations: ca penis, penile cancer.

**Figure 2 medicina-59-01623-f002:**
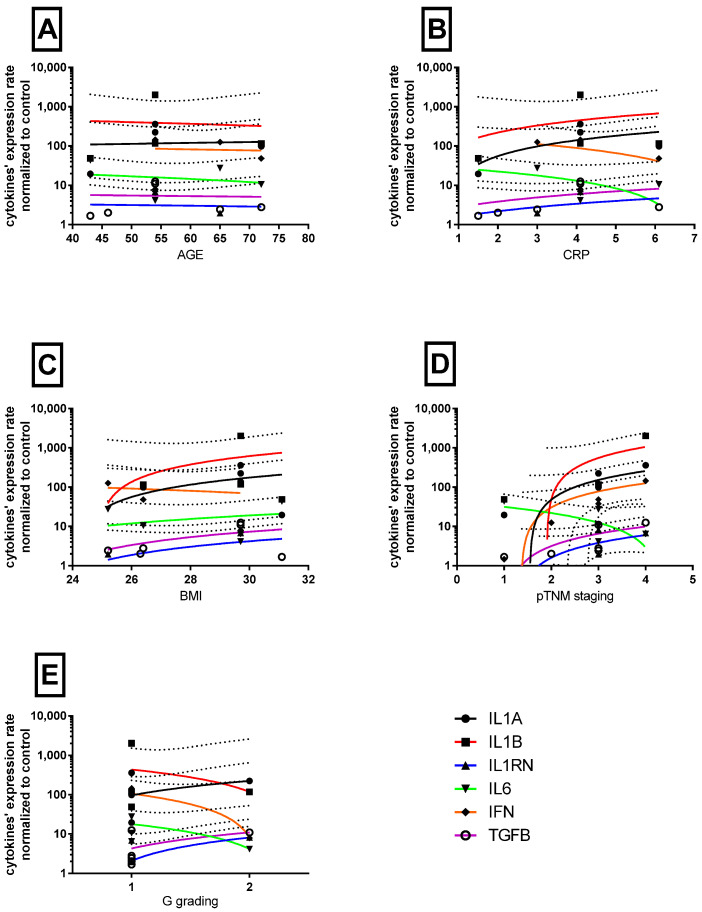
Correlation plots of cytokine levels in tumor samples and clinical data. The ordinate axes are shown on a logarithmic scale. Selected dots represent expression data normalized to control tissues (n = 1), while other values are represented on horizontal axes. Solid color lines represent linear regression curves with 95% confidence band (dotted lines). (**A**–**C**) represent clinical data (age, (**A**), BMI, (**C**)) or values before surgery (CRP, (**B**)), while pTNM (**D**) stages are coded: carcinoma planoepitheliale veruccosum; 1 TiS; 2 pT1aNxMx; 3 pT2N0Mx; 4; (**E**) represents histological G grading.

**Table 1 medicina-59-01623-t001:** Details of qPCR assays.

Gene Name	Primer Sequences	qPCR Reaction Conditions	qPCR Reaction Content
*IL-1α*	5′-TAGGTCAGCACCTTTTAGCTTC 5′-GTATCTCAGGCATCTCCTTCAG	95 °C, 3 min; 45× (95 °C, 5 s; 59 °C, 10 s; 72 °C, 10 s; 75 °C, 10 s—sample reading) Melting curve: 95 °C, 15 s; 60 °C, 1 min; 60 °C → 95 °C reading every 0.3 °C	5 µL AmplifyMe NoRox SybrGreen (with SybrGreen fluorophore) (Blirt, Gdańsk, Poland), 200 nM each primer, Σ 10 µL
*IL-1β*	5′-CCTTAGGGTAGTGCTAAGAGGA 5′-TACAGACACTGCTACTTCTTGC		
*IL-1RN*	5′-GGCACTTGGAGACTTGTATGAA 5′-GAGCTGAAGTCACAGGAAGTAG		
*IL-6*	5′-CACTCACCTCTTCAGAACGAAT 5′-AGGCAAGTCTCCTCATTGAATC		
*INF-γ*	5′-TGGAAAGAGGAGAGTGACAGAA 5′-TATTGCTTTGCGTTGGACATTC		
*TNF-β*	5′-GAGCTGTACCAGAAATACAGCA 5′-AACTCCGGTGACATCAAAAGAT		
*GUSB*	5′-ATGCAGGTGATGGAAGAAGTGGTG 5′-AGAGTTGCTCACAAAGGTCACAGG		

**Table 2 medicina-59-01623-t002:** Demographic characteristic of patients.

Group	N	Age (y): Mean ± SD; Median (Range)	*p*	BMI: Mean ± SD	*p*	CRP (mg/dL): Mean ± SD	*p*
Control	13	31.38 ± 14.80; 24.0 (21–65)	0.0086	28.21 ± 6.96	Ns (0.351)	1.76 ± 2.85	0.01
Ca penis	6	52.83 ± 11.25; 51.0 (40–69)		28.07 ± 2.39		3.47 ± 1.67	

## Data Availability

Not applicable.

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
