# Peer review of "Pro-Inflammatory Cytokine Gene Expression in Penile Cancer: Preliminary Studies"

_medicina, 2023, doi:10.3390/medicina59091623_

Round 1
Reviewer 1 Report
This is a study of the expression levels of cytokines in penile cancer in just 6 patients. The authors findings are interesting but they need to address these concerns regarding the manuscript:
1. The authors have just looked at mRNA levels of these cytokines. This does not give us any information about their protein levels. I know they probably do not have access to the samples any more but just looking at the mRNA levels does not help is decide therapeutic targets. They should perform Western blots if possible.
2. The age , the stage of progression of cancer of these patients is not mentioned at all. The findings in 3.3 need to be represented as a graph/table instead of just a statement.
3. The background is too concise. More information needs to be added about the role of these cytokines in penile cancer.
English is satisfactory.
Author Response
Thank you for an in-depth review of our study. We tried to answer all bothering questions and modified the manuscript according to the suggestions.
- Unfortunately, we are not able to perform Western blots. This is one of the study limitations as we mentioned in paragraphs 225-228.
"The main limitation of this study is the small number of cases. Moreover, we investigated the expression of genes, but not molecules. Future studies are needed for the expression of proinflammatory cytokines and linking its expression with penile cancer progression and patient survival, especially for INF-γ"
- We performed the calculation once more and added Figure 2. Moreover, after consultation with pathomorfologist we divided patients in clinical advancement according to TNM (carcinoma planoepitheliale veruccosum→ TiS→ pT1aNxMx → pT2N0Mx). We found a positive correlation between increasing clinical staging and IFN-γ mRNA levels in tumor tissue.
paragraphs: 30-33
It is the first study which showed increased expression of cytokines such as IL-1A, IL-1B, IL-6, INF-γ and TGF-β in penile cancer with positive correlation between TNM staging and INF-γ levels in tumor samples (rs = 0.672, p = 0.045), which may be associated with immunosuppressive role of a tumour environment.
paragraphs: 161-166
Despite the observed differences in the level of cytokines in the neoplastic tissue, we did not notice any relationship between cytokine expression and age, BMI values or CRP levels, as presented in Fig. 2 A-C. However, we found a positive correlation between increasing clinical staging (carcinoma planoepitheliale veruccosum→ TiS→ pT1aNxMx → pT2N0Mx) and IFN-γ mRNA levels in tumor tissue (Fig. 2 D). There was no association between histopathological grade and cytokines’ expression (Fig. 2 E).
paragraphs: 170-175
Figure 2. Correlation plots of cytokines levels in tumor samples and clinical data. The ordinate axes were shown on a logarithmic scale. Selected dots represent expression data normalized to control tissues (n=1), while other values are represented on horizontal axes. (A, B, C) plots represent clinical data (age, A, BMI, C) or values before surgery (CRP, B), while pTNM (D) stages are coded: carcinoma planoepitheliale veruccosum;1 TiS;2 pT1aNxMx;3 pT2N0Mx; 4; (E) represents histological G grading.
paragraphs: 211-214
Our observation of an increase in the expression of the mRNA of the IFN-γ gene with the clinical advancement of the neoplastic disease may confirm the increased immunosuppressive effect of the stromal and influx cells of the immune system.
paragraphs: 231-234
Expression of cytokines such as IL-1A, IL-1B, IL-6, INF-γ and TGF-β are higher in penile cancer but it is not associated with penile cancer progression and patients survival, while positive correlation between INF-γ levels in cancer samples and clinical advancement may be associated with immunosuppressive role of a tumour environment.
3. We developed the background.
paragraphs: 47-58
The tumor immune microenvironment (TIME) appears to be an important factor in the pathogenesis of penile cancer. According to Chu et al. this very complex microenvironment can provide to penile tumor progression [4]. TIME is divided into three types: immune inflamed, immune desert, and immune excluded [5]. Since phimosis and balanitis are considered risk factors for penile cancer, should penile tumors be recognize as inflamed tumor? The immune-inflamed environment is more favorable for T-cell activation and expansion, including IL-1A, IL-1B, IL-1RN, TGFβ-1 and INF- γ[5]. Moreover, in a recent study, we confirmed the higher expression of NF-κB1 and NF-κB2 (nuclear factor of kappa light polypeptide gene enhancer in B-cells 1 and 2) in penile cancer tissue. NF-κB1 and NF-κB2 take part at stimulate the secretion of proinflammatory cytokine [6]. Additionally, pro-inflammatory cytokines are overexpressed in penile lichen sclerosus which is considered as risk factor for penile cancer [7].
Reviewer 2 Report
The manuscript proposes to uncover and verify the mRNA inflammation panel in penile cancer. Also, it aims to identify to infer if some demographic parameters have an impact on the same panel. The manuscript's structure is clear and straight to the point. In view of improving the quality of the manuscript and proceeding to its acceptance, I recommend that the authors verify some questions I have during the evaluation of the manuscript:
Major topics:
Men with penis cancer had, in the mean, high age. It is widely known that with age, all people tend to enhance the inflammation in their bodies. Thus, did you compare your results to a control group without disease for that age?
What can be the main reasons for the enhancement of the mRNA levels? (Mutations on DNA that lead to overexpression of the gene in a “compensatory mechanism”? ; Exacerbated stimuli for mRNA production?) Please, formulate a hypothesis for your results, based on literature.
Minor topics:
Keywords should be in alphabetic order.
Statistical analysis – remove the number 2 that it has at the beginning of the topic.
I consider the mRNA quantification part important and should be added to the methodology.
Author Response
Thank you for an in-depth review of our study. We tried to answer all bothering questions and modified the manuscript according to the suggestions.
Major topics:
- Our control group is a little bit younger than the penile cancer group. However, in our opinion, the differences are not so huge. The range of age was (21-65) in the control group vs (40-69) in the penile cancer group. To our knowledge, this is the first study about proinflammatory cytokines expression in penile cancer. We haven't opportunity to find the control group at the same age as the penile cancer group.
- Our hypothesis about the higher expression of proinflammatory cytokines expression is the creation of the immune-inflamed environment which is more favourable for T-cell activation and expansion, including IL-1A, IL-1B, IL-1RN, TGFβ-1 and INF- γ
paragraph 47-58
The tumor immune microenvironment (TIME) appears to be an important factor in the pathogenesis of penile cancer. According to Chu et al. this very complex microenvironment can provide to penile tumor progression [4]. TIME is divided into three types: immune inflamed, immune desert, and immune excluded [5]. Since phimosis and balanitis are considered risk factors for penile cancer, should penile tumors be recognize as inflamed tumor? The immune-inflamed environment is more favorable for T-cell activation and expansion, including IL-1A, IL-1B, IL-1RN, TGFβ-1 and INF- γ[5]. Moreover, in a recent study, we confirmed the higher expression of NF-κB1 and NF-κB2 (nuclear factor of kappa light polypeptide gene enhancer in B-cells 1 and 2) in penile cancer tissue. NF-κB1 and NF-κB2 take part at stimulate the secretion of proinflammatory cytokine [6]. Additionally, pro-inflammatory cytokines are overexpressed in penile lichen sclerosus which is considered as risk factor for penile cancer [7].
Minor topics:
- We change the keywords in alphabetic order
- We removed the number 2, before the Statistical analysis
- We added the methodology in paragraphs 97-118 and Table 1
Assessment of the mRNA expression of IL-1A, IL-1B, IL-1RN, IL-6, TGF-β1 and IFN-γ genes
RNA isolation was optimized by adapting a modified method of Chomczynski and Sacchi [8] using a Total RNA Mini protocol isolation kit (A&A Biotechnology, Poland). Briefly, RNA-Later samples were defrosted, drained of liquid with a sterile paper towel; 3×3×3 mm tissue fragment was cut out for RNA extraction. The remaining tissue sample of similar size was placed in a sterile vial and immediately placed in liquid nitrogen, and then was stored at -80°C. The processed biopsy tissue was cut with sterile scissors to as small as possible fragments and placed in a 1.5 ml Eppendorf tube with 800 µl Fenozol. The tube was incubated in TS-100C (BioSan, Latvia) thermoblock at 50°C for 45 min. After adding 200 µl chloroform (POCH), samples were gently mixed, and incubated at room temperature (RT) for 5 min, followed by centrifugation at 12000 rpm for 15 min at 4°C. The next steps of RNA extraction were carried out by the manufacturer’s protocol with the final elution volume of 100 µl RNAse-free water. After RNA quantity and purity assessment (Epoch 800 plate reader), RNA was stored at -80°C for further analyses. cDNA synthesis was performed as previously described [9]. Total RNA samples (2 µg) were reverse transcribed with RevertAid Reverse Transcriptase (Fermentas; Thermo Fischer Scientific, Inc.). Details concerning the qPCR methodology are provided in Table 1. One µl of four-times diluted cDNA was used in 10 µl total volume of qPCR reaction. All reactions were run in duplicate; the measurement of glucuronidase beta (GUSB) gene expression was used for the normalization of qPCR results with Livak and Schmittgen’s 2DDCq method [10,11].
Round 2
Reviewer 1 Report
The authors have addressed my concerns.